# The Impact of Sclerostin Levels on Long-Term Prognosis in Patients Undergoing Coronary Angiography: A Personalized Approach with 9-Year Follow-Up

**DOI:** 10.3390/jpm11030186

**Published:** 2021-03-06

**Authors:** Adam Kern, Tomasz Stompór, Jolanta Kiewisz, Bartłomiej E. Kraziński, Jacek Kiezun, Marta Kiezun, Jerzy Górny, Ewa Sienkiewicz, Leszek Gromadziński, Dariusz Onichimowski, Jacek Bil

**Affiliations:** 1Department of Cardiology and Internal Medicine, Medical Faculty, University of Warmia and Mazury in Olsztyn, 10-561 Olsztyn, Poland; lgol@op.pl; 2Department of Cardiology, Voivodal Specialist Hospital in Olsztyn, 10-561 Olsztyn, Poland; jgorny@wss.olsztyn.pl (J.G.); stewa@mp.pl (E.S.); 3Department of Nephrology, Hypertension and Internal Medicine, Medical Faculty, University of Warmia and Mazury in Olsztyn, 10-561 Olsztyn, Poland; stompin@mp.pl; 4Department of Human Histology and Embryology, Medical Faculty, University of Warmia and Mazury in Olsztyn, 10-561 Olsztyn, Poland; Jolanta.kiewiesz@uwm.edu.pl (J.K.); bartlomiej.krazinski@uwm.edu.pl (B.E.K.); jacek.kiezun@uwm.edu.pl (J.K.); 5Department of Animal Anatomy and Physiology, Faculty of Biology and Biotechnology, University of Warmia and Mazury in Olsztyn, 10-561 Olsztyn, Poland; marta.kiezun@uwm.edu.pl; 6Department of Anaesthesiology and Intensive Care, School of Medicine, Collegium Medicum, University of Warmia and Mazury, Olsztyn, 10-561 Olsztyn, Poland; dariusz.onichimowski@uwm.edu.pl; 7Clinical Department of Anaesthesiology and Intensive Care, Voivodal Specialist Hospital in Olsztyn, 10-561 Olsztyn, Poland; 8Department of Invasive Cardiology, Centre of Postgraduate Medical Education, 02-507 Warsaw, Poland; jacek.bil@cmkp.edu.pl

**Keywords:** coronary artery disease, multivessel disease, bone metabolism, sclerostin, Klotho, osteocyte, cardiovascular events, coronary angiography, myocardial infarction, stroke, death

## Abstract

Sclerostin might play a role in atherosclerosis development. This study aimed to analyze the impact of baseline sclerostin levels on 9-year outcomes in patients without significant renal function impairment and undergoing coronary angiography. The primary study endpoint was the rate of major cardiovascular events (MACE), defined as a combined rate of myocardial infarction (MI), stroke, or death at 9 years. We included 205 patients with a mean age of 62.9 ± 0.6 years and 70.2% male. Median serum sclerostin concentration was 133.22 pg/mL (IQR 64.0–276.17). At 9 years, in the whole population, the rate of MACE was 34.1% (*n* = 70), MI: 11.2% (*n* = 23), stroke: 2.4% (*n* = 5), and death: 20.5% (*n* = 42). In the high sclerostin (>median) group, we observed statistically significant higher rates of MACE and death: 25.2% vs. 43.1% (HR 1.75, 95% CI 1.1–2.10, *p* = 0.02) and 14.6% vs. 26.5% (HR 1.86, 95% CI 1.02–3.41, *p* = 0.049), respectively. Similar relationships were observed in patients with chronic coronary syndrome and SYNTAX 0–22 subgroups. Our results suggest that sclerostin assessment might be useful in risk stratification, and subjects with higher sclerostin levels might have a worse prognosis.

## 1. Introduction

The osteocyte is considered the master cell that governs bone turnover and interactions between bone, the parathyroid gland, the kidneys, and possibly the cardiovascular system [1]. It is a crucial player in the process of mechanotransduction, which changes the mechanical stress applied to the skeleton into biochemical (humoral) signaling with the involvement of several organs that are apparently not directly involved in bone mineral metabolism [2]. Osteocytes linked together in a lacunar–canalicular system are considered to be an active endocrine organ that releases sclerostin, fibroblast growth factor 23 (FGF23), Dickkopf (DKK-1), phosphate-regulating neutral endopeptidase, osteoprotegerin, matrix extracellular phosphoglycoprotein, and osteocalcin [2]. These proteins do not operate only locally but may influence remote organs [3]. They also interact with other hormones or regulatory proteins, such as parathormone (PTH) and Klotho protein, and may be linked with cardiovascular events and other outcomes in subjects with chronic kidney disease (CKD). The crosstalk between bone and the cardiovascular system via mentioned hormones and regulatory proteins is called the bone–kidney–heart axis [4,5].

Sclerostin is a glycoprotein synthesized by osteocytes. Sclerostin regulates bone formation and hampers signaling in the Wnt/β-catenin pathway [6]. The Wnt/β-catenin signaling pathway exerts a key function in the endothelium’s inflammation process, vascular calcification, and mesenchymal stem cell differentiation [7,8]. In consequence, sclerostin is suspected to play a part in atherosclerosis development [9,10].

Unfortunately, not many research papers have analyzed the pathophysiologic relationships between atherosclerosis and sclerostin in subjects without advanced CKD [11,12]. In a previous report, we revealed no evident association between sclerostin levels and coronary artery disease severity. However, sclerostin levels correlated with intact PTH (iPTH) and Klotho protein [13,14]. The present research aimed to analyze the impact of baseline sclerostin levels on 9-year outcomes in subjects without significant renal function impairment and undergoing coronary angiography.

## 2. Materials and Methods

### 2.1. Study Population

Consecutive subjects undergoing coronary angiography between 29 June 2011 and 17 November 2011 and fulfilling the inclusion criteria were enrolled, as described previously [14]. In short, all subjects met the following inclusion criteria: age ≥ 40 years and <80 years, chronic coronary syndrome (CCS) or acute coronary syndrome (ACS) as an indication for coronary angiography, creatinine level prior to the procedure ≤1.2 mg/dL (and estimated glomural filtration rate (eGFR) > 60 mL/min/1.73 m^2^), and left ventricular ejection fraction >30%. The exclusion criteria included a diagnosis of infectious or autoimmune disease or current use of an anti-inflammatory/immunosuppressive treatment, a thyroid disorder, a history of viral hepatitis or liver failure of any origin, significant valvular heart disease, a malignant tumor, or current use of drugs that could influence serum sclerostin levels, as well as bone metabolism. The independent ethics committee of the University of Warmia and Mazury in Olsztyn approved the study protocol. Signed informed consent was obtained from all the participants. Follow-up was performed by phone or by letters. If no contact was obtained, the status (dead or alive) was verified in a national database compiled by the Ministry of Interior and Administration.

### 2.2. Coronary Artery Disease Advancement

Coronary angiography was performed using IOM ARTIS DTC Siemens, with the contrast medium volume ranging between 50 and 375 mL per procedure. All coronary angiograms were evaluated by two independent interventional cardiologists who were blinded against clinical data. Then, coronary artery disease severity was verified, with the SYNTAX score applied by two experienced interventional cardiologists who were not aware of clinical results. We used the SYNTAX score calculator 2.1 (www.syntaxscore.com (accessed on July 2020) to assess all coronary lesions with ≥50% diameter stenosis in an artery with reference diameter >1.5 mm. The coronary artery disease advancement was categorized into three groups: low risk (0–22 points), intermediate risk (23–32 points), and high risk (≥33 points) according to European Society of Cardiology (ESC) guidelines [15].

### 2.3. Clinical Data

At baseline, data on age, sex, and addictions; a full medical history; and medication use were collected through a specific questionnaire. Standardized methods measured height and weight. Body mass index (BMI) was calculated by dividing the weight by the square of the height. Blood pressure was measured according to ESC/European Society of Hypertension (ESH) guidelines [16].

### 2.4. Biochemical Data

Blood samples for analysis were collected on the morning before coronary angiography in planned procedures, and in subjects with ACS, blood samples were taken directly prior to the procedure. We analyzed, among others, the following parameters: total cholesterol, glucose, creatinine, and bone turnover markers (Ca, P, iPTH). eGFR was calculated in accordance with the simplified Modification of Diet in Renal Disease (MDRD) formula. Serum sclerostin levels were measured by the ELISA test (DY1406, R&D SYSTEMS, Minneapolis, MN, USA). The sclerostin detection threshold was 41.5 pg/mL. The FGF-23 intact form was detected at a level of 1.5 pg/mL (60–6600, Immunotopics, Inc., San Diego, CA, USA). The ELISA test sensitivity for Klotho protein was 39 pg/mL (CSB-E13235h, CUSABIO, Wuhan, China). The intra- and inter-assay variability were below 10% for all proteins measured. All blood samples were stored at −80 °C.

### 2.5. Endpoints and Definitions

The primary study endpoint was the rate of major cardiovascular events (MACE), defined as a combined rate of myocardial infarction (MI), stroke, or death at 9 years. The secondary endpoints were the rates of MI, stroke, or death in the whole population as well as in subgroups, i.e., patients with CCS, patients with ACS, and SYNTAX subgroups.

Arterial hypertension was defined based on systolic blood pressure ≥ 140 mmHg, diastolic blood pressure ≥ 90 mmHg, or the use of antihypertensive drugs [16]. Type 2 diabetes mellitus was defined based on a level of HbA1c ≥ 6.5%, a fasting glucose level ≥ 126 mg/dL obtained on two separate days, or the use of anti-diabetic drugs [17]. Peripheral artery disease was diagnosed according to the ESC guidelines [18]. Chronic coronary syndrome and ACS were defined based on ischemic symptoms and clinical guidelines [15,19].

### 2.6. Statistical Analysis

Continuous variables were presented as the mean ± standard error of the mean (SEM). The distributions of continuous variables were tested for normality using the Shapiro–Wilk test. Then, variable distribution was compared between subgroups using an unpaired Student’s *t*-test for comparison of two subgroups or an analysis of variance for comparison of more than two subgroups in the case of normally distributed variables, and the Mann–Whitney *U* or Kruskal–Wallis test were applied otherwise. Categorical data were presented as numbers (%) and were compared using the *χ^2^* test or Fisher’s exact test, as appropriate.

The time-to-event data were analyzed using the Kaplan–Meier estimator of the survival curve, and a log-rank test was used to compare the survival distributions between subgroups. Univariable Cox proportional hazard regression models, unadjusted and adjusted for the potential confounders, were built to quantify the effect of sclerostin levels on the risk of selected outcomes. Demographic characteristics (age and sex) were included in the adjusted models.

The level of statistical significance was 0.05. We applied two-sided tests. Formal sample size calculation was not performed, as the study had an explorative nature, and the number of subjects enrolled was limited by both the number of subjects referred for the coronary angiography and the length of the enrollment period. Statistical analyses were performed using R 3.0.2 for OS (R Foundation, Vienna, Austria) [20].

## 3. Results

### 3.1. Study Population

We included 205 patients with a mean age of 62.9 ± 0.6 years, and males were 70.2% (Figure 1). The most common comorbidities were arterial hypertension (75.1%), dyslipidemia (52.2%), metabolic syndrome (41.5%), and diabetes (36.1%). The median serum sclerostin level was 133.22 pg/mL (IQR 64.0–276.17) (Table 1). Participants were divided into two groups based on their median serum sclerostin value: below/equal to the median (the low sclerostin group) and above the median (the high sclerostin group). As reported earlier, serum sclerostin levels did not differ in men vs. women, patients with hypertension vs. normotensive, diabetic vs. non-diabetic, and those that did or did not suffer from metabolic syndrome (data not shown). Detailed follow-up at 9 years was available in 198 patients (96.6%); however, in 100%, we had data on dead/alive status.

### 3.2. Whole Population Survival Analysis

At 9 years, in the whole population, the rate of MACE was 34.1% (*n* = 70), MI: 11.2% (*n* = 23), stroke: 2.4% (*n* = 5), and death: 20.5% (*n* = 42) (Table 2). In the high sclerostin group, we observed statistically significant higher rates of MACE and death, 25.2% vs. 43.1% (HR 1.75, 95% CI 1.1–2.10, *p* = 0.02) and 14.6% vs. 26.5% (HR 1.86, 95% CI 1.02–3.41, *p* = 0.049), respectively (Figure 2).

### 3.3. Chronic Coronary Syndrome and Acute Coronary Syndrome Subgroups Survival Analysis

In the chronic coronary syndrome subgroup, the MACE rate at 9 years was 32.6% (*n* = 42), MI rate: 10.1% (*n* = 13), stroke rate: 0.8% (*n* = 1), and death rate: 21.7% (*n* = 28) (Table 2). In the high-sclerostin group, we observed statistically significant higher rates of MACE and death, 21.5% vs. 43.8% (HR 1.93, 95% CI 1.05–3.56, *p* = 0.04) and 12.3% vs. 31.3% (HR 2.51, 95% CI 1.19–5.27, *p* = 0.02), respectively (Figure 3).

Interestingly, the MACE rates between the CCS and ACS subgroups did not differ (32.6% vs. 36.8%, *p* = 0.66). In addition, in the ACS subgroup, no statistically significant impact of sclerostin levels was observed. MACE, MI, stroke, and death rates did not differ between subgroups (Table 2, Figure 4).

### 3.4. SYNTAX Subgroups Survival Analysis

The MACE rates did not differ between SYNTAX 0-22, SYNTAX 23-32, and SYNTAX ≥ 33 subgroups (*p* = 0.72). In the SYNTAX 0–22 subgroup, the rates of MACE and death were significantly larger with high sclerostin levels, 23.3% vs. 44.4% (HR 1.82, 95% CI 1.03–3.21, *p* = 0.04) and 12.3% vs. 29.2% (HR 2.26, 95% CI 1.10–4.62, *p* = 0.03), respectively. In the SYNTAX 23–32 subgroup, the rates of MACE, MI, and death were numerically higher in subjects with high sclerostin levels (Table 2, Figure 5).

## 4. Discussion

The main findings of our study are as follows: (1) In the whole population group, the MACE rate at 9 years was 34.1% (*n* = 70), MI rate: 11.2% (*n* = 23), stroke rate: 2.4% (*n* = 5), and death rate: 20.5% (*n* = 42); (2) High sclerostin levels were associated with worse prognosis (MACE and death) in the whole study population, in subjects with chronic coronary syndromes, and subjects with low advancement of coronary artery disease (SYNTAX 0–22); (3) The prognosis of subjects with CCS and ACS did not differ significantly at 9-year follow-up.

We must stress that our population was relatively old (62.9 years), and males dominated (72%). Both these factors could influence the sclerostin levels. However, in our previous study, we did not observe significant differences in terms of age and sex between high- and low-sclerostin subgroups [14]. Our previous paper also showed that baseline sclerostin levels did not differ significantly between subjects with no obstructive disease, one-vessel disease, two-vessel disease, or three-vessel disease/left main stem (*p* = 0.40) [14]. Nevertheless, at a 9-year follow-up, we proved that sclerostin levels virtually affected prognosis in patients with chronic coronary syndrome, but not in patients with acute coronary syndrome. Our study is quite unique due to two reasons. The first is associated with a very long follow-up and 100% of data gathered regarding deaths. The second one tackles the issue of kidney function. Most previous papers focused on sclerostin in patients with moderate/severe chronic kidney disease [21,22].

Gaudio et al. revealed that in ambulatory subjects, serum sclerostin levels correlated positively with carotid intima-media thickness (r = 0.314, *p* = 0.03) and inversely with the augmentation index, i.e., a surrogate for arterial stiffness (r = −0.286, *p* < 0.05). Moreover, sclerostin was an independent predictive factor of pulse wave velocity [23]. Further, other studies assessed the impact of sclerostin levels on carotid atherosclerosis. Morales-Santana et al. observed that in subjects with diabetes type 2, sclerostin levels were strictly associated with increased carotid intima-media thickness (*p* < 0.05), carotid plaques (*p* < 0.01), and aortic calcification (*p* < 0.01) [9]. Kirkpantur et al. also proved that sclerostin levels correlated with carotid intima-media thickness in subjects undergoing hemodialysis [7], whereas Zhao et al. proved this in subjects with CKD stage 3–5 [24]. In a similar study, Chen et al. showed that sclerostin levels were not only linked with carotid artery atherosclerosis but also with all-cause deaths [25]. However, in our study, sclerostin levels did not significantly impact the stroke rate.

Ghardashi-Afousi et al. wrote an interesting paper in which they assessed the impact of high-intensity interval training in subjects with diabetes type 2 on sclerostin levels [26]. Authors proved that such training led to sclerostin level decreases and carotid intima-media thickness reduction. Importantly, VO_2_peak negatively correlated with serum sclerostin levels.

Further, sclerostin levels are thought to be associated with atherosclerosis in other vessel beds. Teng et al. showed that elderly subjects had higher statin use (*p* < 0.05), serum iPTH (*p* < 0.01), C-reactive protein (CRP, *p* < 0.05), and sclerostin levels (*p* < 0.01) in the low ankle-brachial index group than in subjects with a normal ankle-brachial index [11]. In a multivariable logistic analysis, in elderly subjects, sclerostin levels (OR 1.05, 95%CI 1.01–1.09, *p* < 0.01) were the independent predictive factor of peripheral artery disease.

He et al. showed that sclerostin levels were associated with adverse outcomes in elderly subjects with CCS undergoing percutaneous coronary interventions [12]. Surprisingly, in the 3-year observation, the authors revealed that subjects with high sclerostin levels were characterized by a significantly decreased risk of cardiovascular events and more prolonged survival. This contradicts our results, in which lower sclerostin levels were associated with a better prognosis. This might be partially explained by three-fold longer follow-up and better kidney function in our study. Moreover, the disparate correlation between sclerostin levels and cardiovascular outcomes in hemodialysis subjects seems to be confirmed by other studies [27,28]. However, more recent studies showed similar results to ours in patients with CKD [29,30]. Nevertheless, Klingenschmid et al. showed that sclerostin levels were not related to incident cardiovascular events in the general population [31].

In our study, an elevated cardiovascular risk was linked to serum sclerostin concentrations above the median. As proven earlier, higher sclerostin levels are markers of increased vessel calcification (including coronary arteries) as well as an ongoing inflammatory process [32]. In addition, the ongoing inflammation is associated with atherosclerosis development [33]. At baseline, in our study, in subjects undergoing coronary angiography, obstructive coronary artery disease was present in 76% of patients. Higher sclerostin levels were indicators of more progressive atherosclerosis development that leads to increased incidence of cardiovascular events.

There are certain limitations of our study. The first one is that we could not differentiate between cardiac and non-cardiac deaths, and therefore, all deaths were categorized as all-cause deaths. In specific subgroups, the low number of subjects could lead to the lack of statistical significance, especially in SYNTAX 23–32 and SYNTAX ≥ 33 subgroups. Finally, seven subjects lost to follow-up could affect the myocardial infarction and stroke rates.

## 5. Conclusions

Our results suggest that sclerostin assessment might be useful for risk stratification in subjects undergoing coronary angiography without impaired renal function. Subjects with higher sclerostin levels were characterized by a worse prognosis.

## Figures and Tables

**Figure 1 jpm-11-00186-f001:**
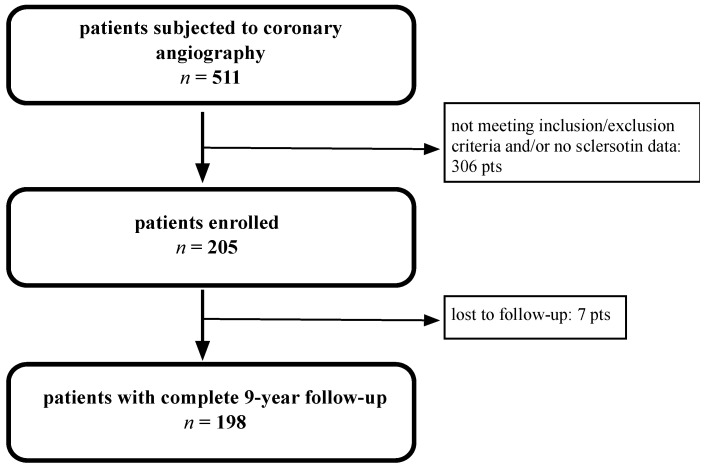
Study flow chart.

**Figure 2 jpm-11-00186-f002:**
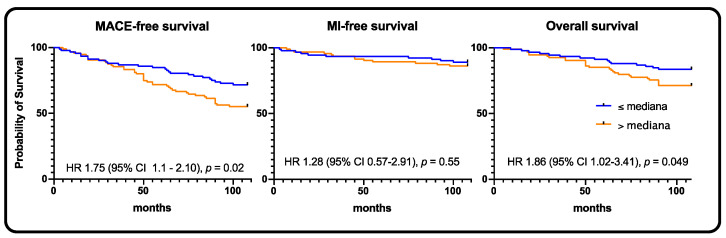
Whole population—Kaplan–Meier curves for MACE, myocardial infarction, and all-cause death, respectively.

**Figure 3 jpm-11-00186-f003:**
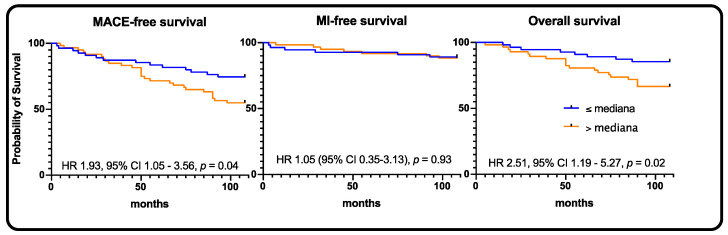
Chronic coronary syndrome population—Kaplan–Meier curves for MACE, myocardial infarction, and all-cause death, respectively.

**Figure 4 jpm-11-00186-f004:**
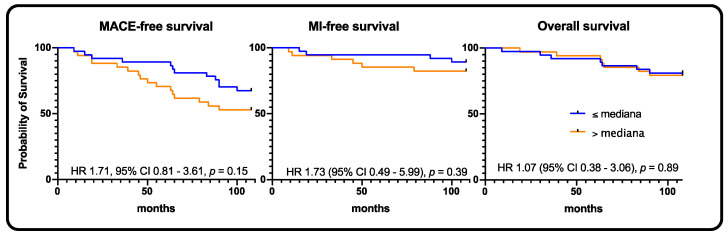
Acute coronary syndrome population—Kaplan–Meier curves for MACE, myocardial infarction, and all-cause death, respectively.

**Figure 5 jpm-11-00186-f005:**
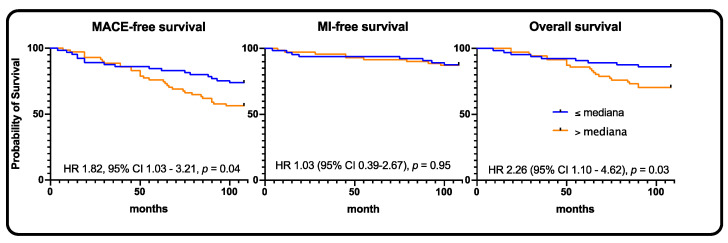
SYNTAX 0–22 population—Kaplan–Meier curves for MACE, myocardial infarction, and all-cause death, respectively.

**Table 1 jpm-11-00186-t001:** Baseline characteristics and laboratory findings.

Parameter	Patients *n* = 205
Baseline characteristics	
male	144 (70.2%)
age (years)	62.9 ± 0.6
BMI (kg/m^2^)	27.6 ± 0.3
arterial hypertension	154 (75.1)
dyslipidemia	107 (52.2)
diabetes	74 (36.1)
metabolic syndrome *	85 (41.5)
chronic coronary syndrome	129 (62.9)
acute coronary syndrome	76 (37.1)
left ventricle ejection fraction (%)	54.8 ± 0.7
SYNTAX score	15.1 ± 0.8
Laboratory findings	
Total cholesterol (mg/dL)	188.9 ± 3.8
Glucose (mg/dL)	117.5 ± 2.8
Creatinine (mg/dL)	0.9 ± 0.01
eGFR (mL/min/1.73 m^2^)	86.5 ± 1.6
Phosphate (mg/dL)	3.15 ± 0.1
Calcium (mg/dL)	9.06 ± 0.04
Ca x P (mg^2^/dL^2^)	28.6 ± 0.5
Sclerostin (pg/mL) **	133.21 (64.0–276.17)
iPTH (pg/mL)	36.1 ± 2.1
KLOTHO (pg/mL)	232.1 ± 15.0
FGF23 (pg/mL)	1.37 ± 0.05

* according to NCEP-ATPIII criteria; ** median and IQR. BMI—body mass index, eGFR—estimated glomural filtration rate; iPTH—intact parathormone; FGF23—fibroblast growth factor 23.

**Table 2 jpm-11-00186-t002:** Adverse event rates at 9 years.

	Whole Group	Low Sclerostin Group	High Sclerostin Group
whole population	*n* = 205	*n* = 103	*n* = 102
MACE	70 (34.1)	26 (25.2)	44 (43.1) *
MI	23 (11.2)	10 (9.7)	13 (12.7)
stroke	5 (2.4)	1 (0.9)	4 (3.9)
death	42 (20.5)	15 (14.6)	27 (26.5) *
chronic coronary syndrome	*n* = 129	*n* = 65	*n* = 64
MACE	42 (32.6)	14 (21.5)	28 (43.8) *
MI	13 (10.1)	6 (9.2)	7 (10.9)
stroke	1 (0.8)	0	1 (1.6)
death	28 (21.7)	8 (12.3)	20 (31.3) *
acute coronary syndrome	*n* = 76	*n* = 39	*n* = 37
MACE	28 (36.8)	12 (30.8)	16 (43.2)
MI	10 (13.2)	4 (10.3)	6 (16.2)
stroke	4 (5.3)	1 (2.6)	3 (8.1)
death	14 (18.4)	7 (17.9)	7 (18.9)
SYNTAX 0–22	*n* = 145	*n* = 73	*n* = 72
MACE	49 (33.8)	17 (23.3)	32 (44.4) *
MI	17 (11.7)	8 (10.9)	9 (12.5)
stroke	2 (13.8)	0	2 (2.8)
death	30 (20.7)	9 (12.3)	21 (29.2) *
SYNTAX 23–32	*n* = 38	*n* = 20	*n* = 18
MACE	14 (36.8)	5 (25)	9 (50)
MI	4 (10.5)	1 (5.0)	3 (16.7)
stroke	2 (5.3)	1 (5.0)	1 (5.6)
death	8 (21.1)	3 (15.0)	5 (27.8)
SYNTAX ≥ 33	*n* = 22	*n* = 12	*n* = 10
MACE	7 (31.8)	4 (33.3)	3 (30)
MI	2 (9.1)	1 (8.3)	1 (10)
stroke	1 (4.5)	0	1 (10)
death	4 (18.2)	3 (25)	1 (10)

* *p* < 0.05; MACE—major adverse cardiovascular event; MI—myocardial infarction.

## Data Availability

The data presented in this study are available on request from the corresponding author.

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
