# Peer review of "The Impact of Sclerostin Levels on Long-Term Prognosis in Patients Undergoing Coronary Angiography: A Personalized Approach with 9-Year Follow-Up"

_jpm, 2021, doi:10.3390/jpm11030186_

Round 1

Reviewer 1 Report

This manuscript is after all well written. The authors investigate the impact of baseline sclerostin levels on 9-year outcomes in patients without significant renal function impairment and undergoing coronary angiography, showing that high sclerostin levels were associated with worse prognosis in terms of major cardiovascular events (MACE) and death in the whole study population, in subjects with chronic coronary syndromes, and subjects with low advancement of coronary artery disease.

The paper is of interest.  The major strength of this study is represented by the long follow-up period, but the entire manuscript needs to be carefully edited by a native English speaker for word usage and grammar.

I suggest you some keywords more pertinent to your article: sclerostin – osteocyte - cardiovascular events - coronary angiography - myocardial infarction – stroke – death

“Introduction” section – lines 37- 38:  This statement needs a reference (e.g.: doi: 10.1210/er.2012-1026)

“Introduction” section – lines 39-41:  This statement needs a reference (e.g.: reference n°1)

“Introduction” section - line 54: you could add some other reference: (e.g. doi: 10.1007/s00198-018-4563-0)

“Study population” section – lines 70-71: please replace ”and a written informed consent form” with the more suitable sentence “Signed informed consent was obtained from all the participants”.

Discussion” section, line 240: please, replace the term with the correct form “revealed”

Have you considered the influence of the predominant gender (males 72%) and the relatively old age (mean age of 62.9±0.6 years) of the sample population as a possible factor influencing the high levels of sclerostin? Please add a comment in the discussion section.

Author Response

This manuscript is after all well written. The authors investigate the impact of baseline sclerostin levels on 9-year outcomes in patients without significant renal function impairment and undergoing coronary angiography, showing that high sclerostin levels were associated with worse prognosis in terms of major cardiovascular events (MACE) and death in the whole study population, in subjects with chronic coronary syndromes, and subjects with low advancement of coronary artery disease. The paper is of interest.  

  1. The major strength of this study is represented by the long follow-up period, but the entire manuscript needs to be carefully edited by a native English speaker for word usage and grammar.

Answer: Thank you for your kind review. We tried to improved manuscript language.

  1. I suggest you some keywords more pertinent to your article: sclerostin – osteocyte - cardiovascular events - coronary angiography - myocardial infarction – stroke – death

Answer: We modified keywords according to the suggestions. Sclerostin was already present.

  1. “Introduction” section – lines 37- 38:  This statement needs a reference (e.g.: doi: 10.1210/er.2012-1026)

Answer: We added the proposed paper by Dallas et al.

  1. “Introduction” section – lines 39-41:  This statement needs a reference (e.g.: reference n°1)

Answer: We added the reference.

  1. “Introduction” section - line 54: you could add some other reference: (e.g. doi: 10.1007/s00198-018-4563-0)

Answer: We added the proposed paper by Muto et al.

  1. “Study population” section – lines 70-71: please replace ”and a written informed consent form” with the more suitable sentence “Signed informed consent was obtained from all the participants”.

Answer: We modified this according to the reviewer’s suggestion.

  1. Discussion” section, line 240: please, replace the term with the correct form “revealed”

Answer: We corrected this typo.

  1. Have you considered the influence of the predominant gender (males 72%) and the relatively old age (mean age of 62.9±0.6 years) of the sample population as a possible factor influencing the high levels of sclerostin? Please add a comment in the discussion section.

Answer: We added a comment in the discussion sections, however, as we showed earlier no significant differences in terms of age or sex were observed between high and low sclerostin subgroup. We added the following comment: “We must stress that our population was relatively old (62.9 years) and males dominated (72%). Both these factors could influence on the sclerostin levels. However, in our previous study we did not observe significant differences in terms of age and sex between high and low sclerostin subgroups [14]”.

Reviewer 2 Report

The topic of the manuscript is very interesting. The manuscript is nicely written and the study is properly performed. Could authors comment on the pathophysiological association of high sclerostin levels and worse prognosis in patients undergoing coronary angiography?

Author Response

The topic of the manuscript is very interesting. The manuscript is nicely written and the study is properly performed. Could authors comment on the pathophysiological association of high sclerostin levels and worse prognosis in patients undergoing coronary angiography?

Answer: Thank you for your kind comment. We added the following pathophysiological explanation: “In our study elevated cardiovascular risk was linked to serum sclerostin concentrations above the median. As proved earlier higher sclerostin levels are markers of increased vessel calcification (including coronary arteries) as well as an ongoing inflammatory process [33]. And the ongoing inflammation is associated with atherosclerosis development [34]. In our study, at baseline in patients undergoing coronary angiography, obstructive coronary artery disease was present in 76% of patients. Higher sclerostin levels were indicators of more progressive atherosclerosis development that lead to increased incidence of cardiovascular events”